# Gaussian Copula Embeddings

**Chien Lu**    **Jaakko Peltonen**
Tampere University

## Abstract

Learning latent vector representations via embedding models has been shown promising in machine learning. However, most of the embedding models are still limited to a single type of observed data. We propose a Gaussian copula embedding model to learn latent vectorial representations of items in a heterogeneous-data setting. The proposed model can effectively incorporate different types of observed data and, at the same time, yield robust embeddings. We demonstrate that the proposed model can effectively learn in many different scenarios, outperforming competing models in modeling quality and task performance.

## 1   Introduction

Representation learning is a prominent machine learning approach for working with originally non-vectorial data. Embedding models learn latent vectorial representations for data items that appear together with a context, through modeling the interactions between each center item and its context items. The approach was first introduced as a language model [14] which learns word representations through modeling the probability of the appearance of a central word given surrounding context words. The word appearance is modeled as an observation from a multinomial word distribution. Building on this notion, exponential family embeddings were proposed [19] which further generalized the original model to a class of models suitable for many observed data types, which have been shown promising in different domains. However, the ability of such models to incorporate heterogeneous data is still limited.

Data in many modern domains is heterogeneous, involving simultaneous observation of different data types such as categorical values, integers and real-valued numbers, and having varied distributions within each data type, hence it is difficult to model them together as observations in vectorial embedding; naive unified solutions that ignore the difference of the data types would not yield good models. In particular, the different data types are often distributed over varied scales and with various distributional shapes: naive normalization coupled to modeling with a single distributional assumption would not suffice to yield robust embedding models, and would leave them vulnerable to extreme values and distributional shapes not corresponding to the assumed ones. An equally pressing problem is how to flexibly model relationships (dependencies) between the several observed variables with their differing distributions: naive modeling strategies ignoring the variable dependencies would again yield poor embedding models.

We solve the mentioned challenges by introducing a novel Gaussian copula based latent representation learning model. The model learns vectorial embedding representations for items leveraging the center-context item interactions, but unlike previous embedding models the proposed model is able to learn embeddings in a setting with multivariate data having heterogeneous data types and distributions.

For computational efficiency, we introduce a set of variational auto-encoder based inference algorithms. In experiments on five different scenarios, the proposed model is shown to be effective, outperforming competitive methods in task-based evaluations and yielding insights in a social media analysis task.

Our contributions are:

- We introduce a general-purpose representation learning framework which incorporates multiple, heterogeneous data. Our work allows embedding models to include multiple data types and distributional assumptions in well-founded probabilistic joint modeling through the Gaussian copula; other approaches such as exponential family embeddings have generalised to different data types but still treat them individually.

- To our knowledge, ours is the first work which brings the advantages of Gaussian copula to learning representation vectors from heterogeneous data. The Gaussian copula is intuitive and has proven effective in machine learning research, thus it is an attractive solution which was neglected in representation learning. We close this gap, and the result shows it handles heterogeneous data well.

- We develop an efficient inference procedure based on semiparametric estimation and variational autoencoder. Previous works have used MCMC for inference and in many such works, lack of scalability has limited their application to larger amounts of data.

- We demonstrate the effectiveness of our model on five different scenarios, each with a real-world data set and corresponding quantitative and qualitative evaluation.

This paper is organized as follows. Section 2 introduces the necessary background notions of embedding models and copula models. Section 3 introduces the Gaussian copula model and Section 4 develops the inference algorithms. Five different scenarios of using the proposed model are provided 5, each evaluated with an experiment on a real-world data set. Section 6 draws the conclusions.

## 2 Background

### 2.1 Embedding models

Learning latent representations based on the interactions between the observed item and its contexts has been an imperative topic in machine learning. It was first introduced as a language model to model relations between words [14]. The framework has been later generalized to model other different co-appearance patterns such as in community embedding [23].

In brief, let the item $i$ and its context $j$ be two items (such as two words, or two communities). The probability of them appearing (D = 1) in the same context (such as in the same sliding window) can be modeled as

$$P(D = 1|i, j) = \left( \frac{1}{1 - e^{-\boldsymbol{\rho}_i^\top \boldsymbol{\rho}_j}} \right) \tag{1}$$

where $\boldsymbol{\rho}_i$ and $\boldsymbol{\rho}_j$ are the embedding vectors of the items. Note that instead of using the same kinds of vectors for both roles, the context items can have their own context vectors $\boldsymbol{\alpha}$, thus the above probability becomes

$$P(D = 1|i, j) = \left( \frac{1}{1 - e^{-\boldsymbol{\rho}_i^\top \boldsymbol{\alpha}_j}} \right) . \tag{2}$$

Exponential Family Embeddings [19] is an extension which has further generalized the model to data beyond text. Let $x_n^{(i)}$ be the value of the item $i$ at the location $n$, which has its context $\mathbf{c}_n$. In exponential family embedding, the value of $x_n$ depends on its context $\mathbf{c}_n$ and is generated from an exponential family distribution

$$x_n^{(i)}|\mathbf{c}_n \sim \mathbf{ExpFam}\left( \eta_n\left( \boldsymbol{x}_{\mathbf{c}_n} \right), t\left( x_n \right) \right) \tag{3}$$

where $\eta_n\left( \boldsymbol{x}_{\mathbf{c}_n} \right)$ is the natural parameter, and $t\left( x_{n,v} \right)$ denotes the sufficient statistics. The natural parameter is modeled as a function of an inner product of the embedding vector $\boldsymbol{\rho}$ and the context vector $\boldsymbol{\alpha}$ so that

$$\eta_n\left( \boldsymbol{x}_{\mathbf{c}_n} \right) = g\left( \boldsymbol{\rho}_i^\top \frac{1}{|\mathbf{c}_n|} \sum_{n' \in \mathbf{c}_n} x_{n'}^{(i')} \boldsymbol{\alpha}_{i'} \right) . \tag{4}$$

As the exponential family can model different observation distributions, the embedding models are no longer limited to modeling co-appearance (binary) observations. It has been applied to different domains such as grouped data [18] and graph data [1].

Negative sampling or sub-sampling is a common practice when training embedding models. The notion is to consider only a randomly generated subset of the items that do not occur at the location $n$. That is, if an item $i$ does not occur at a location, the probability of that negative occurrence $P(D = 0|i, j) = \left( \frac{1}{1-e^{-\rho_i^\top \alpha_j}} \right)$ is integrated into the objective function. In exponential family embeddings, if the item $i$ is generated as a negative sample, the corresponding pseudo observed value is encoded as $x_n^{(i)} = 0$.

## 2.2 Gaussian copula

A $J$-dimensional copula $\mathbb{C}$ is a probability distribution on $[0, 1]^J$ where each of its univariate marginal distributions is a uniform distribution on $[0, 1]$. That is, given a set of uniform distributed random variables $U_1, \ldots, U_J$, a copula is the joint cumulative distribution

$$\mathbb{C}(u_1, \ldots, u_J) = P(U_1 \leq u_1, \ldots, U_J \leq u_J) . \tag{5}$$

The key idea of copula modeling is to use the copula to model the dependencies between several variables having arbitary types and marginal distributions. Let $\mathbf{x}$ be a random vector of length $J$, and let $j \in 1 \ldots J$ index the elements (random variables) in $\mathbf{x}$. According to Sklars' theorem [21], the cumulative distributions (CDFs) of the variables in $\mathbf{x}$ can be modeled by a copula

$$F(x_1, \ldots x_J) = \mathbb{C}\left( F_1(x_1), \ldots, F_J(x_J) \right) \tag{6}$$

where $F$ is the joint CDF and $F_j(x) = P(X_j \leq x)$ is the $j$-th marginal CDF. Since each marginal CDF value is in $[0, 1]$, the right-hand side is a copula regardless of what distributions the individual marginal CDFs have. If every $F_j$ is continuous, then the $\mathbb{C}$ is unique. In this way, the copula encodes the structure of variable dependencies, while allowing each of the variables $x_1 \ldots x_J$ to be of a different type and to have differing kinds of marginal CDFs.

In this paper, we consider a Gaussian copula, which is one of the widely used copula models; we introduce the model to the representation learning task. A Gaussian copula is defined as

$$\mathbb{C}(u_1, \ldots u_J) = \Phi_J\left( \Phi^{-1}(u_1), \ldots, \Phi^{-1}(u_J)|\mathbf{C} \right) \tag{7}$$

where $\Phi_J$ is a $J$-dimensional Gaussian CDF with a correlation matrix $\mathbf{C}$, and $\Phi^{-1}$ is the inverse function of the standard univariate Gaussian CDF. With the Gaussian copula, the joint CDF of observed data can be modeled as

$$F(x_1, \ldots x_J) = \mathbb{C}(F_1(x_1), \ldots F_J(x_J)) = \Phi_J\left( \Phi^{-1}(F_1(x_1)), \ldots, \Phi^{-1}(F_J(x_J))|\mathbf{C} \right) . \tag{8}$$

The Gaussian copula can also be expressed in terms of a latent Gaussian variable representation. First, a latent vector $\mathbf{z}$ is generated from a Gaussian distribution

$$\mathbf{z} \sim \mathbf{N}(0, \Omega) \tag{9}$$

wih covariance matrix $\Omega$ which corresponds to the correlation matrix $\mathbf{C}$ in equation (7). Then for each $j$, the observed data value $x_j$ is obtained from the inverse of the univariate marginal $F_j^{-1}$ according to the generated latent variable $z_j$ so that

$$x_j = F_j^{-1}\left( \Phi\left( \frac{z_j}{\sqrt{\omega_{jj}}} \right) \right) \tag{10}$$

where $\omega_{jj}$ is the $j$-th diagonal element of $\Omega$. In principle, equations (9) and (10) could be used to derive an equation for the likelihood of the data values $x_j$ at each observation, in order to use such likelihoods for parameter fitting. However, instead the practical approach in Gaussian copula research is to derive an *extended likelihood*, and we will do that for the vectorial embedding task.

## 3 Gaussian Copula Embeddings

We now introduce Gaussian Copula Embeddings (GCE), which perform representation learning for data with heterogeneous-type observations of items in contexts. Let $\{\mathbf{x}_1 \ldots \mathbf{x}_N\}$ be the observations.

Each observation $\mathbf{x}_n$ is of a particular item, occurring in a context; moreover, each occurrence of the item is observed with multiple feature values (variables). Let $\mathbf{x}_n = \boldsymbol{x}_n^{(i)}$ denote that the item $i$ occurs at location $n$ and is observed carrying $J$ variables $\boldsymbol{x}_n^{(i)} = [x_{n,1}^{(i)}, \ldots, x_{n,J}^{(i)}]^\top$, where the location $n$ comes with a context $\boldsymbol{c}_n$ which contains a collection of context item indices.

The principle of the GCE model is that the multiple heterogeneous observations will be generated based on the relationship of the central item $i$ to its context, characterized by several embedding vectors, and the dependencies between the observations will be characterized by a Gaussian copula. For each central item $i$ there is an underlying embedding vector $\boldsymbol{\rho}_i \in \mathbb{R}^{K \times 1}$. In addition to occurring as a central item, each item may also occur as part of a context. The context $\boldsymbol{c}_n$ of the location $n$ will contain several items $i'$. The roles of the items in the context will be characterized by context vectors: unlike a traditional embedding model that generates only one type of observation, here each context item $i'$ has a set of variable specific context vectors $\{\boldsymbol{\alpha}_{i',j} | j \in 1, \ldots J\}$.

We develop the GCE model based on the latent representation equations (9) - (10). For each item $i$ the underlying embedding vector $\boldsymbol{\rho}_i \in \mathbb{R}^{K \times 1}$ is generated from a multivariate normal distribution

$$\boldsymbol{\rho}_i \sim \boldsymbol{N}(0, \boldsymbol{I}) . \tag{11}$$

Following the latent variable representation of Gaussian copula, all $J$ observations of an item at a location will be generated based from a latent vector. The latent variable vector $\mathbf{z}_n$ is generated as

$$\boldsymbol{z}_n^{(i)} \sim \boldsymbol{N}(0, \mathbf{I} + \boldsymbol{R}_n \boldsymbol{R}_n^\top) \Longleftrightarrow \boldsymbol{z}_n^{(i)} \sim \boldsymbol{N}(\boldsymbol{R}_n \boldsymbol{\rho}_i, \boldsymbol{I}) \tag{12}$$

where the matrix $\boldsymbol{R}_n \in \mathbb{R}^{K \times J}$ is constructed based on the embedding vectors of the items in the context $\boldsymbol{c}_n$ for all observation variables. For each observation variable $j$, the corresponding column $\boldsymbol{r}_{n,j}$ of the matrix $\boldsymbol{R}_n$ is constructed as

$$\boldsymbol{r}_{n,j} = \frac{1}{|\boldsymbol{c}_n|} \sum_{i' \in \boldsymbol{c}_n} \boldsymbol{\alpha}_{i',j} \tag{13}$$

where $i'$ are the items in the context of the location $n$, $\boldsymbol{c}_n$. Here, for simplicity, we set the prior of $\boldsymbol{\alpha}$ to be a multivariate normal distribution

$$\boldsymbol{\alpha}_{i',j} \sim \mathbf{N}(0, \lambda_\alpha^{-1} \mathbf{I}) \tag{14}$$

with a diagonal covariance matrix where the constant $\lambda_\alpha$ is a precision parameter which controls the constraints on $\boldsymbol{\alpha}$. Using the exchangeability in equation (12), the above generating process can be also written as

$$\boldsymbol{z}_n^{(i)} \sim \boldsymbol{N}(\boldsymbol{\mu}_n^{(i)}, \boldsymbol{I}), \text{ where } \boldsymbol{\mu}_n^{(i)} = [\mu_{n,1}^{(i)}, \ldots, \mu_{n,J}^{(i)}] \text{ and } \mu_{n,j}^{(i)} = \boldsymbol{\rho}_i^\top \frac{1}{|\boldsymbol{c}_n|} \sum_{i' \in \boldsymbol{c}_n} \boldsymbol{\alpha}_{i',j} . \tag{15}$$

The observations are then obtained from the latent variables according to the Gaussian copula equations. The $j$th observed value $x_{n,j}^{(i)}$ is obtained as

$$x_{n,j}^{(i)} = F_j^{-1}\left( \Phi\left( \frac{z_{n,j}^{(i)}}{\sqrt{1 + \sum_{k=1}^K r_{n,j,k}^2}} \right) \right) \tag{16}$$

where $z_{n,j}^{(i)}$ is the $j$th element of the latent vector $\boldsymbol{z}_n^{(i)}$ and $r_{n,j,k}$ is the $k$th dimension of the context representation column $\boldsymbol{r}_{n,j}$, and $F_j^{-1}$ is the inverse CDF of the marginal distribution of variable $j$. Inference of the embedding parameters based on the observations will be done with an extended likelihood approach introduced in the next section.

The major difference between our approach and other embedding models is that the GCE can take heterogenous, multivariate observed data into account. Another difference is that GCE further specializes the roles of embedding vectors $\boldsymbol{\rho}$ and context vectors $\boldsymbol{\alpha}$: in traditional exponential family embeddings their roles can be seen as ambiguous, having the same form and a very similar place in the generative equations. In contrast, here the roles are made distinct: the $\boldsymbol{\rho}$ are used to model a general representation of each item as a central item, and the variable-specific context vectors $\boldsymbol{\alpha}_j$ govern the role of each item as a context item for the multiple variables, telling how the different variables interact with the central item as well as controlling the dependencies between the variables.

# 4 Inference

The task of the inference is to fit the embedding parameters $\boldsymbol{\rho}$ and $\boldsymbol{\alpha}_j$ of all items to the observations. We will use a variational inference approach; a key aspect of it is to replace direct evaluation of the likelihood by an *extended likelihood* approach described next.

## 4.1 Extended Likelihood

As pointed out by [7], the naive inverse of equation (16) during the inference is only recommended when the observed data is continuous and follows an easy-to-invert parametric distribution. Moreover, directly inverting the CDF for discrete variables can lead to pushing the latent variables to extremes and affect the validity of the inference. Therefore, an extended rank likelihood has been proposed for the inference of Gaussian copula.

Consider all observations $x_{n,j}$ of variable $j$ at all locations $n = 1, \ldots, N$, and the corresponding latent variables $z_{n,j}$ where for brevity we drop the item indices $i$; each observation may arise from a different item. Denote the latent variables together by a vector $\mathbf{z}_j$ and observed variables together by vector $\mathbf{x}_j$. Since each $z_{n,j}$ is related to the corresponding observed $x_{n,j}$ by two monotonic functions (a gaussian CDF and an inverse CDF), the rank order of the $z_{n,j}$ is the same as that of the $x_{n,j}$. The vector $\mathbf{z}_j$ is one of a set $\mathbf{D}(\mathbf{x}_j)$ of vectors having that rank order:

$$\mathbf{z}_j \in \mathbf{D}(\mathbf{x}_j) = \left\{ \mathbf{z}_j \in \mathbb{R}^N : x_{n,j} < x_{n',j} \Rightarrow z_{n,j} < z_{n',j} \right\} . \tag{17}$$

The $\mathbf{D}(\mathbf{x}_j)$ is the set of possible $\mathbf{z}_j = (z_{1,j}, \ldots, z_{n,j})$ which preserve the ordering of the observed data [20]. Let $\mathbf{D} = \{ \mathbf{Z} \in \mathbb{R}^{J \times N} : \mathbf{z}_j \in \mathbf{D}(\mathbf{x}_j) \; \forall 1 \leq j \leq J \}$ be the set of possible latent variable combinations, such that the rank order is satisfied for each observed variable $j$. Note that ties may happen in the rank orders, for example when discrete variables yield the same value at multiple observations.

In the Gaussian copula model the observed variables are directly obtained as unique transformations of the latent variables. Because the possible latent variables must satisfy the rank orders of the observations, the rank perservation can be inserted into the full likelihood which can then be factorized as

$$P(\mathbf{X}|\mathbf{C}, F_1, \ldots, F_J) = P(\mathbf{X}, \mathbf{Z} \in \mathbf{D}|\mathbf{C}, F_1, \ldots, F_J)$$
$$= P(\mathbf{Z} \in \mathbf{D}|\mathbf{C}) \times P(\mathbf{X}|\mathbf{Z} \in \mathbf{D}, \mathbf{C}, F_1, \ldots, F_J) . \tag{18}$$

The $P(\mathbf{Z} \in \mathbf{D}|\mathbf{C})$ is then taken as the alternative likelihood. It has been proved [8] that it shares the same information bound as using estimator of the full data. Ranks also bring an advantage of robustness as they are unaffected by precise value and thus are less prone to outliers. Using an extended likelihood based on the rank of observed data is the current state of the art practice for Gaussian copula inference [15, 3].

## 4.2 Amortized variational autoencoder

Most of the previous works taking the extended likelihood use Gibbs sampling for inference [15, 3]. Despite some recent improvement such as [9], the sampling process inevitably must update the latent variables for every location $n$, which makes the computation inefficient when the volume of data grows large. To avoid this inefficiency, we develop a stochastic variational inference procedure exploiting the idea of amortized inference [4] and a variational autoencoder [11].

Since the essence of the estimator using $P(\mathbf{Z} \in \mathbf{D}|\mathbf{C})$ is to keep the ranking of $\mathbf{z}_j$ corresponding to the ranking of $\boldsymbol{x}_j$, we then employ the Plackett-Luce model with ties [24] as an alternative likelihood. Let $r(x_{n,j})$ denote the rank of $x_{n,j}$ and $\boldsymbol{r}(\boldsymbol{x}_j)$ denote the vector of rankings corresponding to $\boldsymbol{x}_j$. We then have

$$p(\boldsymbol{r}(\boldsymbol{x}_j)|\mathbf{z}_j) = \prod_q \left( \frac{e^{z_{n,j}}}{\sum_{n' \in C_q} e^{z_{n',j}}} \right)^{\frac{1}{|A_q|}} \tag{19}$$

where $q$ in the product goes over the rank positions, $A_q = \{n : r(x_{n,j}) = q\}$ denotes the items ranked at position $q$ (there may be more than one due to ties), and $C_q = \{n : r(x_{n,j}) \geq q\}$ are items ranked at $q$ or higher. Note that if there are no ties, equation (19) reduces to the standard

Plackett-Luce distribution. The ties are considered in order to handle discrete data. The likelihood for all variables $j$ is then simply $\prod_j \log p(\boldsymbol{r}(\boldsymbol{x}_j)|\boldsymbol{z}_j)$. Moreover, the likelihood can be optimized using a stochastic procedure with a random subset of each $\boldsymbol{x}_j$.

Let us consider again the $J$ latent variables per location, denoted $\mathbf{z}_n^{(i)}$. To avoid exhaustively updating every $\mathbf{z}_n^{(i)}$, we follow the framework proposed by [11]. According to the "reparameterization trick", equations (15) can be rewritten as

$$\mathbf{z}_n^{(i)} = \boldsymbol{\mu}_n^{(i)} + \boldsymbol{\epsilon}_n^{(i)}, \ \boldsymbol{\epsilon}_n^{(i)} \sim N(0, \mathbf{I}) \ . \tag{20}$$

Therefore, during the stochastic inference, the latent variables $\tilde{\mathbf{z}}$ can be simulated based on the other parameters and the random noise. This will save computation and memory because the $\tilde{\mathbf{z}}$ only need to be simulated at the subset of positions samples in the ongoing mini-batch of optimization. The noise is first sampled from a standard normal distribution and a pseudo variable $\tilde{\mathbf{z}}$ is then generated as

$$\tilde{\mathbf{z}}_n^{(i)} = \hat{\boldsymbol{\mu}}_n^{(i)} + \boldsymbol{\epsilon}_n^{(i)}, \boldsymbol{\epsilon}_n^{(i)} \sim N(0, \mathbf{I}) \tag{21}$$

where $\hat{\boldsymbol{\mu}}_n^{(i)} = \hat{\boldsymbol{\rho}}_i^\top \frac{1}{|\boldsymbol{c}_n|} \sum_{i' \in \boldsymbol{c}_n} \hat{\boldsymbol{\alpha}}_{i',j}$, and $\hat{\boldsymbol{\rho}}$ and $\hat{\boldsymbol{\alpha}}$ are point estimates of the embedding vectors which we are optimizing. The log-likelihood function becomes $\tilde{\mathcal{L}}(\boldsymbol{\rho}, \boldsymbol{\alpha}; \mathbf{x}) = \sum_j \log p(r(\boldsymbol{x}_j)|\tilde{\mathbf{z}}_j)$ and the objective function is computed as

$$\mathcal{F} = \tilde{\mathcal{L}}(\boldsymbol{\rho}, \boldsymbol{\alpha}; \mathbf{x}) + \sum_i \log p(\boldsymbol{\rho_i}) + \sum_{i'} \sum_j \log p(\boldsymbol{\alpha_{i',j}}) \ . \tag{22}$$

In each iteration, the gradients of the log-likelihood with respect to $\hat{\boldsymbol{\rho}}$ and $\hat{\boldsymbol{\alpha}}$ can be simply obtained with $\tilde{\mathbf{z}}$ by chain rule via $\nabla_{\boldsymbol{\rho}} \tilde{\mathcal{L}} = \sum_j \frac{\partial \log p(\boldsymbol{x}_j|\tilde{\mathbf{z}}_j)}{\partial \tilde{\mathbf{z}}_j} \frac{\partial \tilde{\mathbf{z}}_j}{\partial \boldsymbol{\rho}}$, and $\nabla_{\boldsymbol{\alpha}_j} \tilde{\mathcal{L}} = \frac{\partial \log p(\boldsymbol{x}_j|\tilde{\mathbf{z}}_j)}{\partial \tilde{\mathbf{z}}_j} \frac{\partial \tilde{\mathbf{z}}_j}{\partial \boldsymbol{\alpha}_j}$. The $p(\boldsymbol{\rho_i})$ and $p(\boldsymbol{\alpha_{i',j}})$ are set to $\mathbf{N}(0, \mathbf{I})$ and $\mathbf{N}(0, \lambda_\alpha^{-1}\mathbf{I})$ according to equations (11) and (14). The gradients of the log-priors are $\sum_i \frac{\partial \log p(\boldsymbol{\rho})}{\partial \boldsymbol{\rho}_i}$ and $\sum_{i'} \sum_j \frac{\partial \log p(\boldsymbol{\alpha})}{\partial \boldsymbol{\alpha}_{i',j}}$ respectively. The gradient to update $\hat{\boldsymbol{\rho}}$ or $\hat{\boldsymbol{\alpha}}$ with respect to $\mathcal{F}$ is then the sum of the corresponding log-likelihood and log-prior gradients.

The complete stochastic inference procedure is given in Algorithm 1. We optimize the embedding vectors iteratively over epochs. In each epoch data is partitioned randomly into mini-batches, and negative samples are generated for each batch in addition to its positive samples: a negative sample has the same location $n$ and context as a positive sample but a randomly chosen different item $i$, and its observed variable values are all set to zero since the item did not occur at that location. The items for the negative samples are chosen from a distribution proportional to a power of the overall item distribution, as is done in word embedding [14]. In experiments we use $M = 1000$ mini-batches and 5 negative samples for each positive sample. Due to the stochastic partitions, for each epoch the latent variables only need to be simulated at the positions in the mini-batch. The optimization then updates the embedding vectors in each epoch by gradient steps with step sizes chosen by the Adam optimizer.

## 5 Empirical Case Studies

In this section we describe 5 different scenarios of using GCE to model the observed data. The precision parameter $\lambda_\alpha$ is set to 0 corresponding to a very wide prior for $\boldsymbol{\alpha}$. We have also tried another setting with a constrained prior, they yield similar results (see supplementary materials).

### 5.1 Product rating data

**Data.** The Anime rating data[1] is a set of user ratings on anime movies and series collected from myanimelist.net. It contains 17562 different anime rated by 325770 different users. Unlike typical product rating data sets, the data set provides how many episodes (integer) the user had watched when rating the anime (discrete), thus there are multivariate heterogeneous observed variables.

**Modeling.** We evaluate GCE by comparing with two other models on their capability to predict held-out ratings. Compared models are exponential family embeddings (Poisson distribution, p-emb) and Poisson matrix factorization model ([2], Pois-MF). The p-emb is selected as a comparison method

---

[1]From Kaggle, https://www.kaggle.com/datasets/CooperUnion/anime-recommendations-database

**Algorithm 1:** Inference Algorithm

---

**input** :Observations $\{x_1 \dots x_N\}$, Context $\{c_1 \dots c_N\}$, initial learning rate $\xi$
**output :**Point estimates of embedding vectors $\hat{\rho}$
      and context vectors $\hat{\alpha}$
**foreach** *epoch* **do**
    Divide input data into $M$ random partitions.
    Generate negative samples.
    **for** $m \leftarrow 1$ **to** $M$ **do**
        Use the $m$-th batch of the data
        Simulate $\tilde{z}_n$ with (15) for every $n$ in the mini-batch
        Compute gradients $\nabla_{\rho}\mathcal{F} = \sum_j \frac{\partial \log p(x_j|\tilde{z}_j)}{\partial \tilde{z}_j} \frac{\partial \tilde{z}_j}{\partial \rho} + \sum_i \frac{\partial \log p(\rho)}{\partial \rho_i}$,
        $\nabla_{\alpha_j}\mathcal{F} = \frac{\partial \log p(x_j|\tilde{z}_j)}{\partial \tilde{z}_j} \frac{\partial \tilde{z}_j}{\partial \alpha_j} + \sum_{i'} \sum_j \frac{\partial \log p(\alpha)}{\partial \alpha_{i',j}}$
        Update $\rho$ and $\alpha$ with $\rho = \rho - \xi * \nabla_{\rho}\mathcal{F}$, and $\alpha = \alpha - \xi * \nabla_{\alpha}\mathcal{F}$
        $\xi$ is set with Adam[10]
    **end**
**end**

---

Table 1: Left: Held-out MAE for Anime rating. Right: Held-out MAE for Match Records.

| | | | | | Kills | | Deaths | |
|---|---|---|---|---|---|---|---|---|
| Model | K = 50 | K = 100 | | Model | K = 50 | K = 100 | K = 50 | K = 100 |
| Pois-MF | 7.4866 | 7.4872 | | n-emb | 30.0125 | 31.3224 | 29.4128 | 30.6495 |
| p-emb | 3.0857 | 3.0691 | | p-emb | 32.5609 | 32.5409 | 32.5620 | 32.5433 |
| GCE | **1.2170** | **1.2207** | | GCE | **14.0867** | **14.0766** | **13.5245** | **13.5106** |

since our model can been seen as an extension of exponential family embeddings; the Pois-MF is a model for the user-rating scenario and it was also a compared method to p-emb in [19]. For all methods, when training the model, we hold out 10% of the data as the testing data set, and the trained models are used to predict the ratings in the test data. For each anime rated by a specific user, other anime rated by the same user are its context. We train GCE with two variables, anime rating and number watched episodes, whereas p-emb and Pois-MF which can only model one variable are trained to model the anime ratings.

To compute the predictive ratings we use (16) with $z_{n,j}^{(i)}$ set to the means $\mu_{n,j}^{(i)}$ computed from the optimized embedding vectors and with $F_j^{-1}$ computed as the inverse of the empirical CDF in the training data. The held-out mean absolute error (MAE) is used as the performance metric. The results are shown in the Table 1 (Left): our model strongly outperforms p-emb and Pois-MF.

## 5.2 Player modeling in online games

**Data.** The HLTV match record data is collected from HLTV.org and records professional match histories of a multiplayer first-person shooter game Counter-Strike: Global Offensive [22]. We used a web crawler to gather the histories of 34900 matches of 4751 professional players from the website. For each match, we collect the match ID, player ID, and records of each player in the match including number of kills and deaths, again yielding multivariate heterogeneous observations.

**Modeling.** We train GCE to learn representation vectors for each player. The context for the player in each match is the set of other players in the same match. Observed data are the numbers of kills and deaths. We again compare our model to two exponential family embeddings (normal and Poisson) because they are the methodologically closest approaches. We train GCA incorporating the two variables at the same time whereas the exponential family embeddings train the model for each variable separately. The predictions for the variables are done as in Section 5.1. We measure MAE for both variables separately. The results are shown in Table 1 (Right): we strongly outperform the comparison methods.

Table 2: Results for Darknet Traffic Classification

| Model | Precision | Recall | F1 | Accuracy |
|---|---|---|---|---|
| DeepImage [6] | 0.86 | 0.86 | 0.86 | 0.86 |
| Random forest (original features) | 0.8374 | 0.7963 | 0.8117 | 0.8917 |
| GCE (K = 20) + Random forest | **0.8955** | 0.8789 | 0.8846 | 0.9347 |
| GCE (K = 30) + Random forest | 0.8945 | **0.8803** | **0.8851** | **0.9355** |
| GCE (K = 40) + Random forest | 0.8952 | 0.8786 | 0.8844 | 0.9346 |

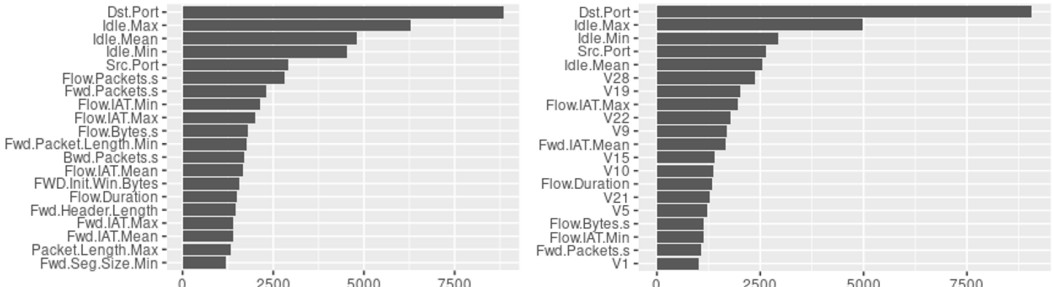

Figure 1: Importance of top-20 variables in the random forest classifiers, x-axis is the mean decrease of Gini impurity, the higher the more important the variable. Left: Variable importances in the random forest for original input variables. Right: The random forest model trained with additional 30 learned features. Learned features V28, V19, and other 7 features are in the 20 most important variables.

### 5.3 Internet traffic classification

**Data.** The CIC Dark-net traffic data set [6] contains 141532 records of darknet traffic. Each record is categorized into a traffic category (Audio-Stream, Browsing, Chat, Email, P2P, Transfer, Video-Stream, and VOIP, 8 categories in total) and contains the source IP, destination IP, and communication observations such as forward and backward bytes, flows, duration, subflow and so on.

**Model.** We use GCE to learn the latent representation for each source IP. In each traffic record, the destination IP is the context of the source IP. The observed data used to train the GCE model are IP co-appearance (Boolean), TCP Flag counts (SYN, RST, PSH, and ACK; integer), and three subflow related measurements (continuous value). We first train the embedding model with different vector dimensions. After training GCE, we incorporate the learned representation vector of each source IP as additional features to the original input variables and train a random forest classifier to predict the traffic category. We compare our model to DeepImage, which is a convolutional neural network based, end-to-end solution proposed in [6]. The results in Table 2 show that the learned additional features not only improve performance of the random forest classifier, but also outperform DeepImage, the state-of-the-art deep learning based classifier. Figure 1 further demonstrates that the learned features play important roles in the classification task.

### 5.4 Graph embedding with node meta-data

**Data.** Spanish Twitch gamers is a subgraph of the Twitch gamers graph data [17]; each node is a Twitch gamer; an edge denotes mutual friendship. The data has 5538 nodes and 85893 edges.

**Model.** We train the embedding vectors incorporating the node-level observations including number of views and life duration. Following the customary graph embedding procedure, we first generate random walks on the graph to simulate a node sequence as input data: 80 walks per node with length 10 steps. Conventionally, limited by the capability of embedding models, most graph embedding models only take appearance of nodes into account. With GCE, we incorporate not only the appearance of nodes but also the views and lifetime of the nodes into the model.

To evaluate our model, we take on link prediction, a classical task for graph embedding models. We hold out 50% of edges randomly into a test set while keeping the remaining training graph connected. In both training and test sets, randomly sampled negative edges are added in equal amount to the

Table 3: Results for Link Prediction: area under the curve (AUC) of the link classification

| Model | Deepwalk | Node2vec | EFGE-bern | EFGE-pois | EFGE-norm | GCE |
|---|---|---|---|---|---|---|
| K = 50 | 0.7151 | 0.7143 | 0.5795 | 0.5934 | 0.6063 | **0.7853** |
| K = 100 | 0.7063 | 0.6612 | 0.5887 | 0.6004 | 0.6291 | **0.7832** |

Table 4: Example GCE model output with the subreddit hyperlink network data. We show the top 4 closest subreddits in terms of the embedding vectors $\rho$ and three different context vectors. The $\alpha_1$ corresponds to the fraction of the characters, $\alpha_2$ corresponds to the fraction of the digits, and $\alpha_7$ corresponds to the semantics.

| | Top 4 closest subreddits | | | |
|---|---|---|---|---|
| Embedding | r/environment | | | |
| $\rho$ | r/cornbreadliberals | r/basicincome | r/energy | r/northcarolina |
| $\alpha_1$ | r/climate | r/green | r/oil | r/water |
| $\alpha_2$ | r/conservation | r/climate | r/likeus | r/tdcs |
| $\alpha_7$ | r/invasivespecies | r/lockcarbon | r/metageopolitics | r/earthdisaster |
| | r/cryptocurrency | | | |
| $\rho$ | r/litecoin | r/noblecoin | r/siacoin | r/bitcoinserious |
| $\alpha_1$ | r/altcoin | r/blackcoin | r/ripple | r/karmacoin |
| $\alpha_2$ | r/cannabis | r/xdp | r/flappycoin | r/litecoin |
| $\alpha_7$ | r/dogenews | r/ethtrader | r/ethdev | r/vos |

positive edges. A logistic regression classifier is trained based on the reduced training graph and the training negative edges, using Hadamard product of embeddings of the edge endpoint nodes as input features; the classifier is used to classify the held-out test-set edges. We compare our model to state-of-the-art, random walk based solutions including Deepwalk [16], Node2vec [5] and Exponential Family Graph Embeddings (EFGE; [1]). The results in the Table 3 show our model outperforms the other competitive models.

### 5.5 Social media community interactions

**Data.** The Reddit Hyperlink Network [12] is a data set of 858488 hyperlinks between 55863 subreddits. For each hyperlink, the data set records the source and destination subreddit, and the description of the hypertext including, e.g., number of words, sentiments, fractions of 5 different character types (i.e., alphabetical, digits, uppercase characters, special characters, white space) and so on.

**Model.** We train the model based on the pairs of source and destination subreddits. The source subreddit in each hyperlink is the context for the destination subreddit. The 5 fractions of different character types, the number of words, and the sentiment are taken as the observed variables. We train a GCE model with $K = 100$.

We demonstrate the closest subreddits based on embedding vectors, and three context vectors. Each reflects a different aspect: take the r/environment for example, when it comes to fraction of alphabetical characters, closest subreddits are related to resources such as r/climate, r/green, and r/oil, and r/water. However, when it comes to sentiment, the closest subreddits for r/environment become r/invasivespecies, r/lockcarbon, r/metageopolitics, and r/earthdisaster.

## 6 Discussions and Conclusions

We introduced Gaussian copula embeddings (GCE), a representation learning model that can incorporate observed data of different data types. A stochastic variational inference algorithm based on semi-parametric estimation for efficient computation is introduced. The empirical case studies demonstrate that our model is effective in many domains outperforming competitive comparison methods, and can provide analytical insights. Moreover, our model can extend the representation learning task to more complex settings and thus bring more opportunities to the research community.

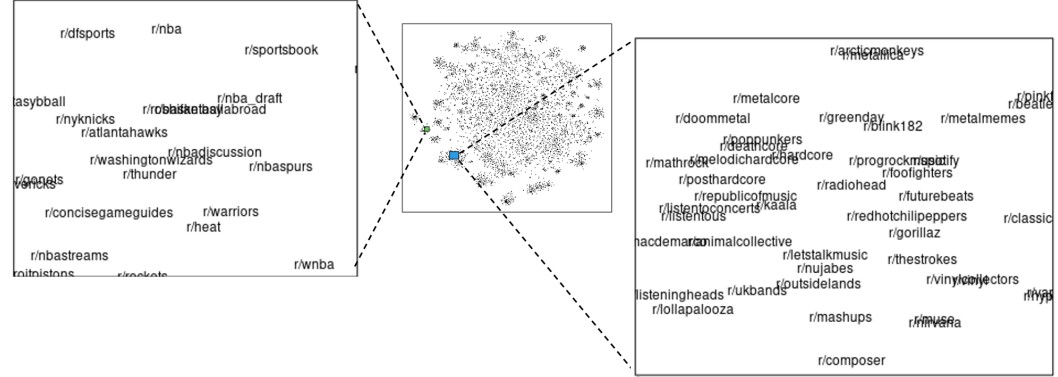

Figure 2: t-SNE visualization of embedding vectors $\rho$. The green area contains the subreddits related to basketball and green area contains subreddits related to music. The learned representation from GCE are semantically meaningful.

In this paper we used a streightforward parametric construction of context and its combination with embedding vectors; however, the GCE framework can be flexibly adapted to other parameterizations, such as integrating it as a layer within deep learning architectures, and integrating context selection mechanisms such as [13]. Our method brings a new way of analyzing data through vectorial embedding which has the potential to bring greater understanding of several phenomena; as usual such tools must be used responsibly to avoid negative societal impact.

## Acknowledgement

This work is supported by the Academy of Finland decisions 312395 and 327352.

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
