# Supplementary material for Gaussian Copula Embeddings

## 1 General settings

**Software**. Our model is implemented in the R language. The computational heavy components are written in C++ integrated with Rcpp and RcppArmadillo packages. The impelmentation can be found in `https://github.com/hummmblelu/GCE`.

**Computing resource** The programs are run on a CPU machine with 1 core and the the memory of 40GB. During the computation, no parallel computation nor GPU resources are needed.

## 2 Error bars in Section 5

In this section we provide more details related to the Section 5 in the main paper.

**Product rating data** (Corresponding to Section 5.1 in the main paper). As mentioned in the main paper, We use 10% held-out data as the testing data set and rest 90% of the data as the training set. We repeat the process 10 times. The mean and standard error (in the parentheses) can be found in Table 1.

**Player modeling in online games** (Corresponding to Section 5.2 in the main paper). Similar to the previous example, the held-out likelihood is employed as the evaluation metric. The mean value of MAE and standard error (in the parentheses) can be found in Table 2.

**Internet traffic classification** (Corresponding to Section 5.3 in the main paper). In this supplementary material, we provide evaluation details over 8 different classes. The detailed evaluation metric with standard deviation over different folds (the the parentheses) are in Table 3

## 3 Experiments with constrained prior

This section provides experiments results with $\lambda_\alpha = 0.01$. Other settings are the same as stated in the main paper. Results corresponding to Section 5.1, 5.2 and 5.4 in the main paper can be found in Table 4. Results corresponding to Section 5.3 can be found in Table 5.

Table 1: Held-out MAE for Anime rating with standard error

| Model | K = 50 | K = 100 |
|---|---|---|
| p-emb | 3.0857 (0.0109) | 3.0691 (0.0170) |
| Pois-MF | 7.4866 (0.0083) | 7.4872 (0.0086) |
| GCE | **1.2170 (0.0060)** | **1.2207** (0.0058) |

Table 2: Held-out MAE for Match Records

| Model | Kills | | Deaths | |
|---|---|---|---|---|
| | K = 50 | K = 100 | K = 50 | K = 100 |
| n-emb | 30.0125 (0.0910) | 31.3224 (0.9076) | 29.4128 (0.0678) | 30.6495 (0.8934) |
| p-emb | 32.5609 (0.0575) | 32.5409 (0.4511) | 32.5620 (0.0596) | 32.5433 (0.4435) |
| GCE | **14.0867** (0.0293) | **14.0766** (0.0422) | **13.5245** (0.0256) | **13.5106** (0.0410) |

Table 3: Detailed Results for Darknet Traffic Classification

| Class | Precision | Recall | F1 | Accuracy |
|---|---|---|---|---|
| Random Forest | | | | |
| Audio-streaming | 0.8966 (0.0063) | 0.9217 (0.0094) | 0.9089 (0.0029) | 0.9530 (0.0042) |
| Browsing | 0.9296 (0.0056) | 0.9827 (0.0022) | 0.9554 (0.0035) | 0.9801 (0.0017) |
| Chat | 0.7212 (0.0213) | 0.8526 (0.0134) | 0.7811 (0.0088) | 0.9117 (0.0053) |
| Email | 0.7919 (0.0360) | 0.5628 (0.0347) | 0.6565 (0.0118) | 0.7780 (0.0164) |
| File-transfer | 0.9108 (0.0080) | 0.8380 (0.0064) | 0.8729 (0.0053) | 0.9155 (0.0032) |
| P2p | 0.9933 (0.0004) | 0.9988 (0.0002) | 0.9960 (0.0003) | 0.9976 (0.0002) |
| Video-Streaming | 0.8307 (0.0145) | 0.7087 (0.0140) | 0.7647 (0.0074) | 0.8490 (0.0066) |
| Voip | 0.6253 (0.0281) | 0.5056 (0.0314) | 0.5579 (0.0110) | 0.7489 (0.015) |
| GCE (K=20) + Random Forest | | | | |
| Audio-streaming | 0.9094 (0.0042) | 0.9376 (0.0055) | 0.9233 (0.0026) | 0.962 (0.0026) |
| Browsing | 0.9338 (0.0017) | 0.9843 (0.0016) | 0.9584 (0.0011) | 0.9816 (0.0008) |
| Chat | 0.8452 (0.0094) | 0.8753 (0.0067) | 0.8599 (0.0035) | 0.9306 (0.0030) |
| Email | 0.8980 (0.0107) | 0.7400 (0.0158) | 0.8113 (0.0107) | 0.8681 (0.0079) |
| P2p | 0.9282 (0.0091) | 0.8632 (0.0061) | 0.8945 (0.0048) | 0.9287 (0.0030) |
| Video-Streaming | 0.9959 (0.0004) | 0.9991 (0.0004) | 0.9975 (0.0002) | 0.9985 (0.0002) |
| File-transfer | 0.8624 (0.0142) | 0.7230 (0.0120) | 0.7865 (0.0080) | 0.8572 (0.0058) |
| Voip | 0.7907 (0.0142) | 0.9083 (0.0189) | 0.8452 (0.0063) | 0.9510 (0.0092) |
| GCE (K=30) + Random Forest | | | | |
| Audio-streaming | 0.9092 (0.0057) | 0.9378 (0.0041) | 0.9233 (0.0036) | 0.9621 (0.0021) |
| Browsing | 0.9362 (0.0037) | 0.9840 (0.0023) | 0.9595 (0.0018) | 0.9819 (0.0011) |
| Chat | 0.8474 (0.0121) | 0.8679 (0.0085) | 0.8575 (0.0070) | 0.9270 (0.0042) |
| Email | 0.8804 (0.0164) | 0.7467 (0.0302) | 0.8076 (0.0163) | 0.8710 (0.0148) |
| File-transfer | 0.9301 (0.0069) | 0.8672 (0.0060) | 0.8975 (0.0049) | 0.9308 (0.0030) |
| P2p | 0.9961 (0.0006) | 0.9992 (0.0003) | 0.9977 (0.0002) | 0.9986 (0.0001) |
| Video-Streaming | 0.8615 (0.0086) | 0.7277 (0.0121) | 0.7889 (0.0076) | 0.8595 (0.0059) |
| Voip | 0.7949 (0.0184) | 0.9117 (0.0165) | 0.8491 (0.0109) | 0.9528 (0.0081) |
| GCE (K=40) + Random Forest | | | | |
| Audio-streaming | 0.9057 (0.0028) | 0.9406 (0.0017) | 0.9228 (0.0012) | 0.9631 (0.0007) |
| Browsing | 0.9358 (0.0024) | 0.9841 (0.0008) | 0.9593 (0.0009) | 0.9818 (0.0001) |
| Chat | 0.8443 (0.0090) | 0.8730 (0.0113) | 0.8584 (0.0085) | 0.9294 (0.0058) |
| Email | 0.8915 (0.0115) | 0.7433 (0.0185) | 0.8105 (0.0088) | 0.8696 (0.0091) |
| File-transfer | 0.9265 (0.0073) | 0.8658 (0.0049) | 0.8951 (0.0041) | 0.9300 (0.0024) |
| P2p | 0.9959 (0.0009) | 0.9991 (0.0006) | 0.9975 (0.0007) | 0.9985 (0.0005) |
| Video-Streaming | 0.8688 (0.0086) | 0.7176 (0.0053) | 0.7860 (0.0025) | 0.8548 (0.0024) |
| Voip | 0.7928 (0.0213) | 0.9055 (0.0068) | 0.8453 (0.0126) | 0.9497 (0.0034) |

Table 4: Left: Held-out MAE for Anime rating. Center: Results for Link Prediction: area under the curve (AUC) of the link classification. Right: Held-out MAE for Match Records.

| Ratings | | AUC | | Kills | | Deaths | |
|---|---|---|---|---|---|---|---|
| K = 50 | K = 100 | K = 50 | K = 100 | K = 50 | K = 100 | K = 50 | K = 100 |
| 1.2176 | 1.2211 | 0.7804 | 0.7817 | 14.0857 | 14.0552 | 13.52452 | 13.5409 |

Table 5: Results for Darknet Traffic Classification

| Model | Precision | Recall | F1 | Accuracy |
|---|---|---|---|---|
| GCE (K = 20) + Random forest | 0.8958 | 0.8797 | 0.8852 | 0.9352 |
| GCE (K = 30) + Random forest | 0.8963 | 0.8808 | 0.8861 | 0.9358 |
| GCE (K = 40) + Random forest | 0.8956 | 0.8792 | 0.8847 | 0.9349 |