# OpenReview forum: "Gaussian Copula Embeddings"
_NeurIPS.cc/2022/Conference — NeurIPS 2022 Accept_

### Official Review · Reviewer_SEhr · 2022-07-05

**Rating:** 7
**Confidence:** 3
**Soundness:** 3 good
**Presentation:** 3 good
**Contribution:** 3 good

**Summary:**

The authors present a model for embedding discrete items such that they model the distribution of multiple heterogeneously distributed context dependent observations. This method employs latent variables that are distributed as multivariate Gaussians, and use Gaussian copulas to capture the dependence between dimensions. The paper presents a key insight that these Gaussian copula models can be used as part of an embedding approach. The paper relates this to previous embedding approaches including word2vec and the more general exponential family embeddings. The paper then presents the model and an inference approach based on the extended likelihood. There are a series of experiments on a wide variety of domains and the results show that the proposed method significantly outperforms comparator methods for each of these.

**Questions:**

For the experimental benchmarks:
Are these the best existing methods, or ones with readily available implementations, or what?

Also, Graph embedding with node meta-data is missing key information. How is the data interpreted as item predictions with context?

On the authors construction of context: are you able to relate this notion of context to other papers/models. Context doesn't just appear in the papers cited but does appear quite widely elsewhere in deep learning and beyond. I wonder if the contribution here is potentially more widely relevant.

I am not sure whether $r_{n,j}$ Equation (13) is as general as it could be? This relates to Equation (4) but I am left wondering what the meaning of making context averaged in this way is, particularly in the context of the authors' own model. For instance, in the HLTV data (Sec. 5.2) the presence of either one of two particularly lethal players in a game may greatly reduce other players scores, but the presence of both players may not have "double the effect" (it may be more or less than that). Could the authors maybe comment on the justification and/or limitations of this?

**Limitations:**

Right at the end of the paper the authors state:
> [A]s usual such tools must be used responsibly to avoid negative societal impact.

Perhaps they could indicate what the concerns are here a little more explicitly.

**Strengths And Weaknesses:**

One significant characteristic of the presented model is that context is first represented as a collection of items (discrete entities) each of which carries its own context embedding vector for each observable dimension of the data. There has been a number of deep learning models over recent years that represent the context of observations in some form or another (as is discussed to some degree in the paper) but this seems to me to be a particularly neat and flexible way. Although the authors do relate this context approach to word2vec, node2vec and Exponential Family embeddings, the paper could do more to relate more generally to other (deep learning) models that employ a notion of context. I am not sure precisely how this relates to what has gone before but I would be interested to know.

The authors run a series of experimental evaluations on different data. The details of these are mostly well described, but some justification for the choice of comparator methods/benchmarks could be given. There does appear to be some missing information (see Questions).

Nonetheless, this is a really interesting approach and, to my mind, represents a significant advance in flexiblity for embedding models. The experiments appear to be well designed, test a broad array of domains and the performance improvements with respect to existing methods are convincing. I believe this is a strong paper and will be of interest to a number of communities, not just those directly interested in embedding.

---

> ### Author Response · Authors · 2022-08-01
> **To reviewer SEhr**
>
> Experiments:
> Yes, they are the best implementations of comparable methods that are readily available. In general, we select one-layer representation learning approaches as the comparison methods. We detail the method selection below for each paper section.
>
> 5.1: Our work can be seen as an extension of EFE, therefore the Poisson embedding is selected. Besides, Poisson matrix factorization is selected due to the recommendation system scenario and it was also compared to in the EFE paper.
>
> 5.2: We compare to EFE for it is the methodologically closest comparison method to our model that is applicable to the data setting. The comparison can demonstrate two key differences between GCE and EFE: GCE takes heterogeneous data and GCE yields robust output.
>
> 5.3: The DeepImage is compared to since it is the method used in the paper where we cite the dataset. DeepImage is a convolutional neural network based, end-to-end solution. The GCE + Randomforest result shows that high-quality features yielded by a simpler representation learning with a classical classifier can outperform such a complex framework.
>
> 5.4: We choose unsupervised, one-layer graph embedding approaches including deepwalk, node2vec, and three types of EFGE. GNN is not considered since it is a multi-layer approach with a different context construction. Note that it is also not considered a comparison method in, for example, the EFGE paper.
>
> 5.5: This section is a demonstration of qualitative evaluation of our approach. The results show that our model reflects the features of the selected variables, e.g., each $\alpha$ reflects a specific aspect of the  environment and cryptocurrency domains.
>
> Graph embedding with node meta-data: The Twitch gamers are regarded as nodes and edges are which gamers have friended each other. We perform random walks along this gamer graph. The nodes are the items and their occurrences their appearances within random walks. For each occurrence in each random walk, the surrounding nodes in the random walk sequence (to the left and to the right) are the context. For each occurrence, the features of a node (Twitch gamer) are the binary indicator of the appearance in that position, number of views of the gamer, and lifetime duration of the gamer.
>
> Context construction:
> In this paper we use a simpler context construction approach as shown in equation (4). Which is inline with works such as EFE, deepwalk, node2vec, and EFGE. The choice is also due to that purpose that we aim at providing a general solution. A more complex context construction can potentially improve the model performance for specific cases.
>
> Generality of eq. (13): We agree that the average can be a simplified way of modeling the effect of the contexts. The current context construction has already yielded good results, nevertheless, combining a more sophisticated modeling mechanism (e.g., [1]) is a promising future direction. We will clarify this in the paper.
>
> Potentially Negative Social Impact:
> The output of representation learning may be used as a decision basis by human experts or automated systems, and they must be used carefully to avoid harmful effects such as:
> 1) Learned representations may not perfectly characterize all properties of the high-dimensional data and should not be used as the sole basis of decision-making for human subjects.
> 2) Representations learned with biased data collection could amplify the disadvantages and bias towards the under-represented groups in downstream use of the embeddings.
> 3) Our model allows easier incorporation of multiple variables in learning. An environmental challenge is that including more variables may yield more computational power consumption. The efficient algorithm developed in the paper is trying to ease this potential impact, nevertheless, power consumption should be monitored in deployment.
>
> [1] Liu, L., Ruiz, F., Athey, S., & Blei, D. (2017). Context selection for embedding models. Advances in Neural Information Processing Systems, 30.

---

### Official Review · Reviewer_jaJh · 2022-07-11

**Rating:** 7
**Confidence:** 4
**Soundness:** 4 excellent
**Presentation:** 3 good
**Contribution:** 3 good

**Summary:**

This paper proposes using Gaussian copula to learn the embeddings under heterogeneous data setting. A rank likelihood is proposed to maintain the rank correspondence of the observed variables and the latent variables. The evaluations on heterogeneous data are sufficient to show the efficacy of the proposed methods.

**Questions:**

1. Why there is no KL term in the training objective of variational auto-encoder? If there is only log-likelihood, probably it should not be called ''variational'', just ''amortized inference''.

**Limitations:**

There is not negative societal impact.

**Strengths And Weaknesses:**

Strengths:
1. The idea of using Gaussian copula for embedding learning is novel and interesting. As there are heterogeneous data types in real worlds, the Gaussian copula could help to naturally capture the underlining correlations with one model. This embedding learning mathematical tool could be broadly influential to various application domains.
2. The author also propose an order preserving likelihood and the corresponding variational auto encoder, for efficient and effective inference.

Weaknesses:
1. It seems there is no ablation study in the evaluations. As the model contains three parts: copula, order likelihood and amortized inference, it is expected to have a thorough ablation to show what improvement each part brings.

------
The author addressed my concerns.

---

> ### Author Response · Authors · 2022-08-01
> **To reviewer jaJh**
>
> Ablation studies / Improvements brought by each model part: The inference part could experiment as an ablation study, but effectively MCMC would just be much slower. The two other key parts (rank likelihood and copula) are essential mathematical components that are not easily 'turned off, but we detail their effect below. The improvements from each model part are:
> - Amortized inference: the amortized inference (Plackett-Luce approach) enabled the computationally efficient stochastic gradient descent as opposed to MCMC which would need to go through every data point in each iteration.
> - Extended rank-likelihood: it makes the model insensitive to data scale (and overall prior scale, see the answer about autoencoder below)
> - Copula: this is the central part that brings the ability to connect the data of different types, without copula our model would not consider variable dependencies.
>
> Variational auto-encoder:
> The KL term of the variational objective function corresponds to the log prior probabilities of $\rho$ and $\alpha$. They have usual Gaussian prior terms; we omitted them from eq. 21 - apologies for the omission, we will fix the equation to include the priors.
>
> However, it turns out that the inference is not sensitive to scales of $\rho$ and $\alpha$, and hence to the variances of their Gaussian priors, for two reasons: firstly, in the likelihood term only the inner product of $\rho$ and the context (weighted sum of the $\alpha$) matters, hence any overall scaling factor is exchangeable between $\rho$ and $\alpha$; secondly, the extended rank likelihood is insensitive to scales of the generated x (as it is based on ranks of the z), hence it is insensitive to the overall scale of rho and $\alpha$ which affects the overall means of z through mu.
>
> In experiments of the paper, we used a very wide prior of $\alpha$; we have also tried with a more constrained prior for both $\rho$ and $\alpha$, results are nearly the same and outperform the comparisons like before (summary below). We will note this in the paper.
>
> GCE results for experiments with more constrained priors (standard deviations of Gaussian priors: $\sigma_{\rho}$ = 1, $\sigma_{\alpha}$ = 10):
>
> Anime rating (Held-out MAE; compare to Table 1 left) \
> K=50 1.217604 \
> K=100 1.221142
>
> HLTV (Held-out MAE; compare to Table 1 right) \
> K=50 kill 14.08569 \
> K=50 death 13.52452 \
> K=100 kill 14.05519 \
> K=100 death 13.54091
>
> Darknet (Precision, Recall, F1, Accuracy; compare to Table 2) \
> K=20 0.8958369    0.8796890   0.8852497   0.9351636 \
> K=30 0.8963159    0.8807927   0.8860507   0.9357741 \
> K=40 0.8956067    0.8791975   0.8846813   0.9349249
>
> Link prediction (AUC; compare to Table 3) \
> K=50 0.7804 \
> K=100 0.7817

---

### Official Review · Reviewer_pvbn · 2022-07-13

**Rating:** 5
**Confidence:** 3
**Soundness:** 3 good
**Presentation:** 3 good
**Contribution:** 3 good

**Summary:**

The authors propose a Gaussian copula embedding model to learn latent vector representations of items in a heterogeneous data setting. The proposed model can effectively incorporate different types of observed data and, at the same time, yield robust embeddings. In the experiments, the authors show the proposed model can effectively learn in many different scenarios, outperforming competing models in modeling quality and task performance.


**Questions:**

What are the advantages of the GCE  in the comparison with deep neural nets-based embedding methods?

**Limitations:**


Experimental comparison with neural network-based methods, e.g. paper[1] and ELMO,  are needed to make the paper stronger.

[1]Vashishth, Shikhar, et al. "Incorporating syntactic and semantic information in word embeddings using graph convolutional networks." arXiv preprint arXiv:1809.04283 (2018).





**Strengths And Weaknesses:**

Strengths:


1. The paper is well written with good structures.

2. The flow of the technical sections is clear and easy to follow.

3. Extensive experiments have been conducted to validate the method.



Weakness:

1. The contribution of the paper is insignificant. The authors apply Gaussian Copula to embedding problems and try to use it to capture interactions in heterogeneous data, and the idea is simple and direct.

2. The authors did not provide a deep explanation of why the Gaussian copula is better than other approaches.  Other embedding methods can also be used for heterogeneous datasets.  I believe by leveraging deep neural nets many existing methods can also learn the interactions in heterogeneous datasets.

=====

The authors addressed some of the concerns and clarified the benefits of the proposed method.

However, the technical and theoretical contributions of the paper are not significant.

I increase the score by 1.

---

> ### Author Response · Authors · 2022-08-01
> **To reviewer pvbn**
>
> Significance:
> 1) To our knowledge, GCE is the first work which brings the advantages of Gaussian copula to learning representation vectors from heterogeneous data.
>
> 2) The Gaussian copula is intuitive and has proven effective in machine learning research, thus it is an attractive solution which was neglected in representation learning. We close this gap, and the result shows it handles heterogeneous data well.
>
> 3) A novel combination of extended rank-based, amortized, stochastic inference is developed in our paper for the Gaussian Copula Model. Previous works have used MCMC for inference and their inefficiency and lack of scalability have limited their application to larger amounts of data.
>
> 4) The key advantage is that GCE advances embedding models to be able to include multiple data types and distributional assumptions in well-founded probabilistic joint modeling (through the Gaussian copula). Other approaches such as exponential family embeddings have generalised to different data types but still treat them individually.
>
> Relations to deep network models and language models:
> 1) To our knowledge, in the representation learning phase of a deep network model, multi-layered structures or aggregation layers such as pooling are common practices when handling heterogeneous data. Despite the effectiveness of such structures, the above-mentioned approach does not yield immediate modeling of interactions directly in a layer of the network, rather they arise only in comparatively opaque ways through weighted neuron sums in further layers.
>
> 2) The compared method DeepImage in section 5.3 is a deep learning model, and we have outperformed it by simply combining GCE and random forest. DeepImage was chosen because it was the solution proposed in the paper where we cite the data (Dark-net). Comparison to deep learning models in other application domains (e.g., language [1], computer vision) is indeed interesting, we leave it for our future work.
>
> 3) Our proposed approach can be further integrated into multi-layer deep model frameworks as the representation learning layer (e.g., as one layer of GNNs; our model has flexibility in that it can be integrated into different parts of such networks). Moreover, it can be used with a variety of context construction options. Applications can include computer vision and language models. Integrating our approach into multi-layer language models (e.g. for context learning) and comparison to other multi-layer NLP works is therefore a potential direction of future work.
>
> 4) In this work, we used the most common way to define the context, i.e., surrounding entities, and so on. The proposed method can fit into more complex usages with more complex context formations.
>
> [1] Vashishth, S., Bhandari, M., Yadav, P., Rai, P., Bhattacharyya, C., & Talukdar, P. Incorporating Syntactic and Semantic Information in Word Embeddings using Graph Convolutional Networks. ACL 2019.

---

> > ### Comment · Reviewer_pvbn · 2022-08-07
> > **Update**
> >
> > Concerns are addressed. The score will be updated.

---

### Author Response · Authors · 2022-08-01
**General response**

We thank all reviewers for the comments, encouragement, appreciation, and suggestions for improving our paper. We briefly highlight the key notions and potentials of our work below. We also respond to each reviewer specifically in separate responses.

Key notions:
1) We focus on developing a general-purpose representation learning model.

2) The solution we provide is of a minimalist nature, requiring only one layer of representation learning. This has two potential advantages. Firstly, the training results can be reused to benefit other downstream machine learning applications. Secondly, it offers the flexibility of being integrated into a more complex, end-to-end solution such as GNN and its derivatives.

3) When selecting the comparison methods, we try to focus on comparable unsupervised, one-layered representation learning approaches such as EFE, deepwalk, node2vec, and EFGE.

Future potential:
1) The developed inference algorithm is efficient and robust. The Gaussian copula + extended rank-likelihood stochastic inference can mediate the effect of outliers (see section 5.1) and is scalable. Besides integrating the proposed method in other models, the modeling and optimization strategies used in developing the method can be further applied in other works for robust modeling.

2) Making sense of data with embedding vectors having clear roles. In this work, we differentiated the functionalities of the embedding vectors $\rho$ and context vectors $\alpha$ for each input variable.

---

### Meta-Review · Area_Chair_a6yh · 2022-08-26

**Recommendation:** Accept
**Confidence:** Less certain

**Metareview:**


This paper was a difficult case, where numerical review scores might be in disagreement with several statements during the discussion phase, and also with some of my comments below.
Let me explain this in more detail.
All reviewers mentioned several strong points, such as:
- the general novelty of the concept of using copulas for learning embeddings
- the overall soundness and clarity of writing
- interesting conceptual parts, for instance the proposed ordering-preserving likelihood
- good experimental validation

On the other hand, also several negative points have been raised by the reviewers, and some of them remained valid after the rebuttal. To me, the most important of these negative points is the statement by one of the reviewers that "the technical and theoretical contributions of the paper are not significant".

During the discussion phase, we discussed several points that were listed as "strengths" in the original reviews, such as the statement in one of the reviews that "the Gaussian copula could help to naturally capture the underlining correlations with one model." My counter-argument was that a Gaussian copula model can capture the underlying dependencies if and only if the true underlying "pure" dependency after removing the marginal effects (i.e. the copula) was indeed Gaussian. But this might be rare in practice, and an essential part of the copula literature deals with non-Gaussian dependency models that can explain practically relevant phenomena such as tail dependency etc..
In the end, we somewhat agreed that the usefulness of Gaussian copulas for this purpose might be not so clear, and that some ablation studies might be needed in order to better understand the contribution of the Gaussian copula to the overall method.

A second question I raised during the discussions concerned the efficiency of the proposed inference method, which had an important role in the paper, since the authors explicitly motivated their work by efficiency problems in "traditional" Gaussian copula estimation procedure. If that really is a main contribution of this paper, I would argue that there should have been a comparison with the Hamiltonian MCMC method for Gaussian copula inference in (Alfredo Kalaitzis & Ricardo Silva, NIPS 2013), which - to my experience - is indeed quite efficient in practice, and this argument was also supported by one of the reviewers.

In the end, to me as an area chair,  this is one of the classical "borderline" cases, where a paper does certainly contain some interesting aspects, but on the other hand, there are also many potential problems and limitations. Since all reviewers finally saw this paper above the threshold, I also recommend to accept this paper, but I still would like to mention that I am not fully convinced about this recommendation.

**Award:**

No

---

### Decision · Program_Chairs · 2022-09-14

Accept